# A Dataset and Benchmark for Copyright Protection from Text-to-Image Diffusion Models

## Abstract

Copyright is a legal right that grants creators the exclusive authority to reproduce, distribute, and profit from their creative works. However, the recent advancements in text-to-image generation techniques have posed significant challenges to copyright protection, as these methods have facilitated the learning of unauthorized content, artistic creations, and portraits, which are subsequently utilized to generate and disseminate uncontrolled content. Especially, the use of stable diffusion, an emerging model for text-to-image generation, poses an increased risk of unauthorized copyright infringement and distribution. Currently, there is a lack of systematic studies evaluating the potential correlation between content generated by stable diffusion and those under copyright protection. Conducting such studies faces several challenges, including i) the intrinsic ambiguity related to copyright infringement in text-to-image models, ii) the absence of a comprehensive large-scale dataset, and iii) the lack of standardized metrics for defining copyright infringement. This work provides the first large-scale standardized dataset and benchmark on copyright protection. Specifically, we propose a pipeline to coordinate CLIP, ChatGPT, and diffusion models to generate a dataset that contains anchor images, corresponding prompts, and images generated by text-to-image models, reflecting the potential abuses of copyright. Furthermore, we explore a suite of evaluation metrics to judge the effectiveness of copyright protection methods. The proposed dataset, benchmark library, and evaluation metrics will be open-sourced to facilitate future research and application. The dataset can be accessed here.

## 1 Introduction

Text-to-image generative models have recently emerged as a significant topic in computer vision, demonstrating remarkable results in the area of generative modeling. These models bridge the gap between language and visual content by generating realistic images from textual descriptions. However, rapid advancements in text-to-image generation techniques have raised concerns about copyright protection, particularly unauthorized reproduction of content, artistic creations, and portraits. A specific concern arises from the use of Stable Diffusion, a state-of-the-art text-conditional latent diffusion model, which has sparked global discussions on copyright, privacy, and safety. Currently, systematic studies evaluating the potential correlation between content generated by Stable Diffusion models and copyright infringement are lacking. Firstly, the inherent ambiguity in defining copyright infringement for text-to-image generative models complicates assessments. Additionally, the absence of a large-scale inference dataset and standardized metrics for defining copyright infringement poses additional challenges in conducting comprehensive studies. In this work, we present the first large-scale standardized dataset focused on copyright protection. We also introduce our evaluation metrics to assess the effectiveness of copyright protection methods.

To begin with, establishing a coherent definition of what content generated by text-to-image generative models can be classified as copyright infringement is imperative. For this study, we focus on infringement within two-dimensional artistic works. We believe that a unique painting style of an artist, virtual representations in artistic creations, and individual portraits all represent forms of creative expression deserving of legal protection. In order to identify instances of infringement in these contexts, it is essential to conduct a holistic analysis of both the technical and semantic components of the created content. This involves examining the fundamental aspects of painting,

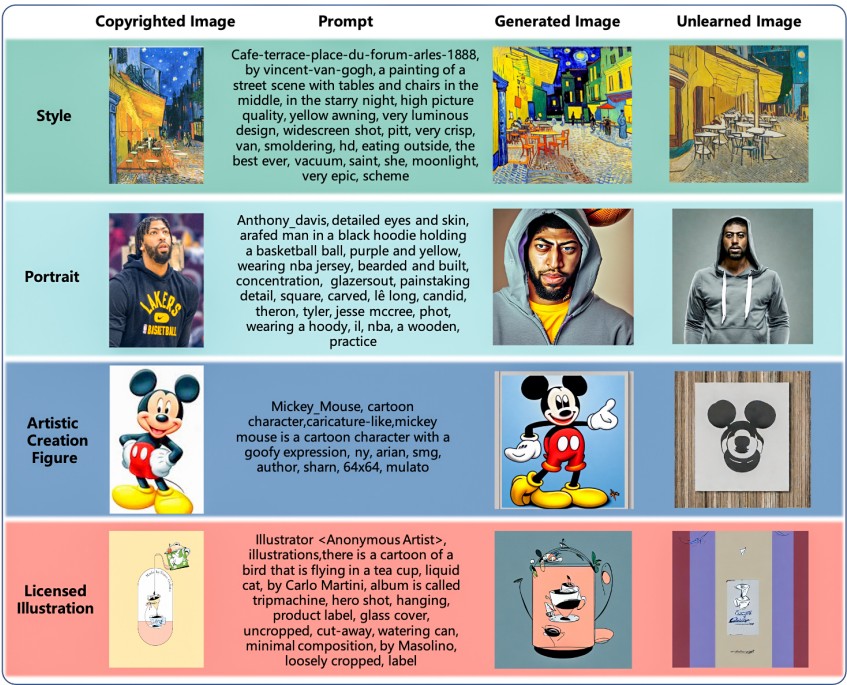

Figure 1: Examples of CPDM dataset composition and unlearned results for copyright protection. The dataset primarily comprises four categories: Style, Portrait, Artistic Creation Figure, and Licensed Illustration. The illustrations are exclusively provided and authorized by the Anonymous Artist, while the remaining images are sourced from WikiArt, WikiArtPedia, and the Internet. The unlearned images are obtained using the methods described in the benchmark 5.

such as brushstrokes, color palettes, lighting effects, and composition, as well as delving into the semantic information within the work.

To validate the effectiveness of our evaluation metrics and to facilitate fair comparisons among various unlearning methods, we introduce a novel dataset: Copyright Protection from Diffusion Model (CPDM) dataset. In the dataset, we have curated four categories of image data involving copyright protection: "Style of Artist" includes images showcasing the unique artistic style of various artists, encompassing distinctive brushstrokes, lines, colors, and compositions found in their paintings. "Portrait of Celebrity" comprises of portrait images of notable individuals. "Artistic Creation Figure" consists of virtual cartoon and character images found in artistic and literary works. Lastly, "Licensed Illustration" encompasses artist illustrations that are protected by copyright.

Furthermore, we conducted extensive benchmark testing on the proposed Copyright Protection from Diffusion Model (CPDM) dataset. In our experiments, we utilized gradient ascent-based and response-based pruning methods for unlearning, specifically targeting the Stable Diffusion model. We assessed performance changes from the unlearning process using the widely-accepted FID (Fréchet Inception Distance) metric and our proposed CPDM metric. This evaluation provided insights into the model's adaptability and its capability to reduce potential copyright infringement risks. Our benchmark offers a straightforward assessment of various methods, eliminating the need for repeated training.

## 2 BACKGROUND AND RELATED WORK

**Text-to-image Generative Models** Recently text-to-image diffusion models Rombach et al. (2022b) have emerged as a crucial research area attracting wide attention. These state-of-the-art methodsNichol et al. (2022); Rombach et al. (2022b); Saharia et al. (2022); Ramesh et al. (2022); Balaji et al. (2023) have exhibited remarkable capabilities in transforming textual information into visually coherent and realistic images, often demonstrating high performance in terms of accuracy. The advancements in these techniques have opened up a plethora of possibilities for a wide range of downstream tasks, such as image editingKawar et al. (2023); Goel et al. (2023); Couairon et al. (2023);

Avrahami et al. (2022), image denoisingHo et al. (2020); Xie et al. (2023) and super-resolutionLi et al. (2021); Gao et al. (2023), etc. The advancements in text-to-image generative models have significantly impacted various industries. However, they also pose challenges to copyright protection. As these models become increasingly adept at creating high-quality images, distinguishing between original artworks and generated ones becomes more complex. This convergence raises critical questions about authorship, intellectual property rights, and the implications for plagiarism in the digital era. Addressing these concerns is imperative.

**Model Unlearning** Carlini et al. (2023) highlights that the privacy of diffusion models is significantly lower compared to generative adversarial networks (GANs) Goodfellow et al. (2020). Under the diffusion framework, models tend to retain certain images from the training data, potentially generating outputs that closely resemble the original images. To remove explicit artwork from large models, Gandikota et al. (2023) presents a fine-tuning method for concept removal from diffusion models. Additionally, Zhang et al. (2023a) presents the "forget me not" method, which enables the targeted removal of specific objects and content from large models within a span of 30 seconds, while minimizing the impact on other content. Somepalli et al. (2023) explores whether diffusion models create unique artworks or directly replicate certain content from the training dataset during image generation. Furthermore, there exist numerous model unlearning methods in the context of image-related tasks, as evidenced by Bourtoule et al. (2021); Ginart et al. (2019); Guo et al. (2019); Graves et al. (2021); Huang et al. (2021), among others. Regarding the comprehensive review of model unlearning, it has provided an overview of unlearning algorithms in various domains such as images, tables, text, sentences, and graphs Zhang et al. (2023b); Nguyen et al. (2022). Although machine unlearning is designed to protect the privacy of target samples, the paper Chen et al. (2021) has demonstrated, in the context of model classification tasks, that machine unlearning might leave traces. Consequently, it is imperative to consider these risks when developing unlearning algorithms for text-to-image generation models.

**Metrics for Images Similarity** Accurately measuring image similarity has always been a challenge yet to be perfectly addressed. For image denoising and super-resolution tasks, researchers have introduced evaluation metrics such as Peak Signal-to-Noise Ratio (PSNR), and Structural Similarity Index Measure (SSIM) to assess the fine-grained similarity between two images Wang et al. (2004). However, these metrics are limited in their ability to evaluate only near-identical images and lack the capacity to assess higher-level similarities. For specific generative tasks, Fréchet Inception Distance (FID) Heusel et al. (2017) has become a prevalent metric, spanning from GAN models to diffusion methods. Nonetheless, FID is designed primarily to evaluate the distance between two sets of images, typically using realistic images as reference points. Consequently, it offers limited insight when dealing with two specific images and their inherent similarity.

**Works in Artistic Image Communities** With the emergence of painting capabilities in models like Stable Diffusion, there has been a growing surge in activity and attention within communities focused on image copyright protection. For instance, websites such as https://stablediffusion.fr/artists and https://www.urania.ai/top-sd-artists have gained prominence. These platforms have curated collections of images that are stylistically similar to the work of over a thousand artists, both contemporary and classical. These artists' styles can be imitated using Stable Diffusion models. In comparison to the efforts of these image communities, our approach involves the collection of real, valuable, and specific images from the art world to be used as training examples for image generation models. This represents a more rigorous form of style imitation. In contrast, the artistic images in the provided links tend to focus on capturing certain aspects of an artist's style, fitting into a broader category of style imitation. For details, see the supplementary materials on page 5.

**Policy, Legal, and Social Impact** The increasing global popularity of AIGC highlights the importance of privacy and copyright issues. AI companies, including OpenAI, have taken measures to address concerns related to data security. The US has proposed establishing a new government agency responsible for approving large-scale AI models. Furthermore, the Chinese Cyberspace Administration has published a document emphasizing AIGC security issues (http:gov). Recently enacted legislation, such as the General Data Protection Regulation (GDPR, https://gdpr-info.eu/) in the European Union, the California Consumer Privacy Act (CCPA, https://oag.ca.gov/privacy/ccpa) in California, and the Personal Information Protection and Electronic Documents Act (PIPEDA, https://laws-lois.justice.gc.ca/ENG/ACTS/P-8.6/index.html) in Canada, have legally solidified this right Zhang et al. (2023b); Chen et al. (2021).

## 3 CPDM METRICS

Determining whether two works constitute plagiarism has long been a pressing issue in both the arts and legal domains. Perceptual evaluation metrics based on the human visual system, such as LPIPS Zhang et al. (2018), have achieved evaluation results that align more closely with human perception. By employing deep feature measurements to assess image similarity, these metrics produce perceptually accurate evaluations. Besides, in the realm of videos, perceptual evaluation metrics like VMAF Sheikh & Bovik (2006) combine human visual modeling with machine learning techniques to achieve impressive results. However, previous research has been limited in addressing this challenge, particularly when measuring the similarity between anchor and generated images.

Both statements highlight the current research gap in copyright protection and the definition of similarity, underscoring the need for further investigation. Then, we collaborated with an artist, Anonymous Artist, who is currently active in the art industry and is notably distinguished in the field of illustration. This artist offered invaluable insights from a professional standpoint to help assess whether two images constitute plagiarism, considering elements such as brushstrokes, color palettes, lighting effects, and composition. For a thorough analysis, we divide these measurements into different categories: semantic and stylistic components. Our objective is to develop a formula that combines both components and provides a scalar metric to quantify the similarity between two images.

**Semantic Metric** We leverage the CLIPRadford et al. (2021) model to generate the semantic embedding, and calculate the metrics by:

$$emb_{ori} = CLIP(Image_{ori}), \qquad emb_{gen} = CLIP(Image_{gen}) \tag{1}$$

$$Loss_{sem} = MSE(emb_{ori}, emb_{gen}) \tag{2}$$

where $Image_{ori}$ and $Image_{gen}$ denote the anchor image and generated image respectively; $CLIP$ denotes the CLIP encoder. In previous studies, cosine similarity has been predominantly employed as the evaluation metric, while, in this research, we utilize Mean Squared Error (MSE) instead. This decision is primarily motivated by two factors: first, the range of the MSE is significantly broader than that of cosine similarity, which makes it easier to observe changes resulting from unlearning; second, the adoption of MSE aligns better with the subsequent style metrics discussed below.

**Style Metric** It's relatively more difficult to measure the similarity in style, in this part, Inspired from the method inGatys et al. (2015), we use the activation output by the CNN networks to calculate the features correlations given by the Gram matrix, in our work we leverage the InceptionV3Szegedy et al. (2015), following the Fréchet Inception Distance metric:

$$G^l = Gram(Inception(Image, l)), \qquad D^l = MSE(G^l_{ori}, G^l_{gen}) \tag{3}$$

$$Loss_{style} = \sum_{i=1}^{n} w^l D^l \tag{4}$$

where $Inception(Image, l)$ denotes passsing the $Image$ through an Inception network and extracting the feature maps from layer $l$. The Gram matrix is then computed to provide a style representation of the image at layer $l$. Furthermore, the dissimilarity between the original and generated images in each layer is represented by MSE of the Gram matrices in each corresponding layer. The total style metric, as described above, is determined by weighting factors $w^l$, which represent the contribution of each layer to the overall style metrics. In our work, $n$ is set to 4, because there are four stages of the InceptionV3 model. The values of parameters $w^l$ needs to be fine-tuned according to the distribution of the images. We provide concrete values applicable to our dataset under Fig. 3. Finally, we denote the total metric as:

$$CPDM = (Loss_{sem} \times Loss_{style})^2 \tag{5}$$

Here, we adopt the squared term to emphasize the significant change before and after the unlearning process as well. And the effectiveness of this quantifiable metrics will be verified in 6.2

Semantic metrics encapsulate the primary content information of images. Irrespective of the image's stylistic attributes, its principal subject is captured within the semantic metric. Furthermore, style metrics encompass attributes, including brushstroke depth, line thickness, color schemes, and compositional elements. These style metrics effectively represent a diverse array of image categories.

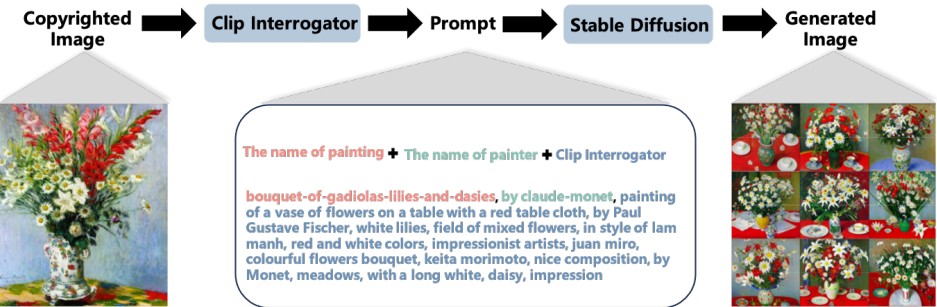

Figure 2: Pipeline for the CPDM Dataset Creation. The Clip Interrogator is utilized to convert copyrighted images into corresponding textual information. This text is subsequently refined and transformed into prompts, which are then inputted into a diffusion model to generate the corresponding infringing images.

## 4 CPDM DATASET

### 4.1 PIPELINE FOR DATASET CREATION

We propose a pipeline to coordinate CLIP, ChatGPT, and diffusion models to generate a dataset that contains anchor images, corresponding prompts, and images generated by text-to-image models, reflecting the potential abuses of copyright. Initially, we collect a set of images that potentially contain copyrighted content, which serves as anchor images. Subsequently, these images are fed into the CLIP-interrogator, allowing us to obtain prompts that correspond to each anchor image. Finally, the prompts are used as input for the stable diffusion model, resulting in the generation of images by the stable diffusion model. Through manual comparisons, we assess whether there is evidence of copyright infringement in terms of style and semantics between the anchor images and the generated images. Ultimately, the anchor images, their corresponding prompts, and the images generated by the stable diffusion model constitute the core components of our dataset. This carefully curated dataset allows for a comprehensive examination of the copyright-related characteristics of the generated content, facilitating rigorous evaluations and analyses of the performance of various techniques in detecting and addressing copyright infringement.

### 4.2 COMPOSITION OF THE DATASET

**Style** Painting artworks often embody the distinctive style of the artist, encompassing aspects such as brushstrokes, lines, colors, and compositions. This artistic style is also a form of copyright that requires protection. WikiArt is an online user-editable visual art encyclopedia, a source from which numerous art-related datasets Karayev et al. (2014); Liao et al. (2022) have been curated. WikiArt already features some 250,000 artworks by 3,000 artists, localized on 8 languages. These artworks are in museums, universities, town halls, and other civic buildings of more than 100 countries. We selected approximately 1500 artworks from 100 artists on WikiArt as the source of anchor images for prompt generation and corresponding content generation using stable diffusion.

**Portrait** The right of portrait refers to an individual's control and use of their own portrait, including facial features, image, and posture. The purpose of publicity rights is to safeguard an individual's privacy, personal dignity, and image integrity, preventing unauthorized use, disclosure, or alteration of their portrait. This legal protection aims to preserve an individual's control over how their likeness is portrayed and commercialized. We utilized web scraping techniques to collect over 200 portrait images from Wikipedia, which is a free, web-based, multilingual encyclopedia that contains articles on a wide range of topics.

**Artistic Creation Figure** Artistic creations, including characters from animations and cartoons, are often protected by law. In this context, we refer to this category as "artistic creation figures." Similar to portraits, we have curated a dataset of 200 influential animated characters and figures by collecting information from reputable sources such as Wikipedia.

Table 1: Statistics and Details of the CPDM dataset.

| Name | Source | Num. of anchor image | Num. of generation |
|---|---|---|---|
| Style | From WikiArt | $\sim$1500 | $\sim$13500 |
| Portrait | From Wikipedia | 200 | 1800 |
| Artistic Creation Figure | From Wikipedia | 200 | 1800 |
| Licensed Illustration | From Anonymous Artist | 200 | 1800 |

**Licensed Illustration** We have obtained authorization to use a portion of Anonymous Artist's artworks in this study. Therefore, we can use his/her illustrations as part of the training dataset for fine-tuning stable diffusion, which will be utilized for the process of simulating infringing artistic paintings.

## 5 CPDM BENCHMARK

**Gradient Ascent-based Approach** To make diffusion models forget a specific copyrighted image, a simple and effective method is to train the model using gradient ascent optimization on that image. For a single image, forgetting can be achieved by optimizing for a few epochs with an appropriate learning rate. More specifically, for a diffusion model with its set of weight parameters $\theta$, to forget the image $Y$ and its corresponding prompt $X$, we update $\theta$ each epoch in the following way:

$$\theta = \theta + \eta \nabla_Y L_{mse}(\theta, X, Y)$$

where $\eta$ is the learning rate and $L_{mse}(\theta, X, Y)$ refers to the loss computed between the generated output using prompt $X$ and the targeted image $Y$. It is evident that the use of gradient ascent optimization has a certain impact on the generative model's capability, even though we only optimize for a small number of epochs.

**Weight Pruning-based Approach** Weight pruning Frankle & Carbin (2018); Han et al. (2015a;b); Liu et al. (2018) is an effective method for reducing a model's parameter count, commonly utilized in model deployment and practical applications. Moreover, this method can be adapted to modify model parameters, enabling the model to forget specific copyrighted images. Inspired by magnitude pruning Han et al. (2015a;b), the core idea behind parameter pruning for forgetting certain infringing images is to mask the weights in the model so that those weights exhibit the strongest response in generating those images. In our experiments, we first feed the image to be forgotten into Stable Diffusion for forward propagation, simultaneously obtaining the gradients of each layer in the network. For the pruning stage, we regard each layer as an individual pruning group. The highest $p_c\%$ of activation values are identified within each layer, and we locate the weights correlated with these values. Subsequently, based on the magnitudes of the gradients of these weights, we set the top $p_w\%$ of weights to zero. This process can be described by the following equation:

$$\theta^* = optim\{\theta | p_c * W_{ij}, \nabla_W L_{ij}, p_c * |Y_{ij}|\}$$

To illustrate, for a layer expressed as $Y = WX$ where $W$ represents the weight (bias term omitted for simplicity), we first select $W_i$ corresponding to the greatest $p_c\%$ of $|Y_{ij}|$, where $|\cdot|$ represents the absolute value operator, $optim\{\cdot\}$ represents updating parameters. Then, for each $W_i$, we prune $p_w\%$ of the elements $W_{ij}$ corresponding to the highest $m\%$ of its gradient values $\nabla_W L_{ij}$, setting these to zero.

## 6 EXPERIMENTS

In the Experiments, we systematically analyze the effectiveness of the proposed CPDM metrics, and provide the results and analysis of benchmark methods on our proposed CPDM datasets.

### 6.1 EXPERIMENTAL SETTING

We conducted experiments on our proposed dataset using two baseline methods. Specifically, we performed image forgetting experiments on the Wikiart, Portrait, Cartoon, and Illustration domains.

We calculated the corresponding metrics to evaluate the effectiveness of image forgetting and used the FID (Fréchet Inception Distance) Dowson & Landau (1982); Heusel et al. (2017) to assess the change in the generative capability of the models after unlearning. In the following two unlearning algorithms, we only made parameter adjustments for the UNet structure of Stable Diffusion Rombach et al. (2022a). We froze the parameters of the text embedding module and the autoencoder module. The respective experiments were conducted on stable diffusion (SD-v1.4), finetuned diffusion model (SD-finetuned), and (SD-v2.1). We performed unlearning experiments on each image in the CPDM dataset. For calculating FID for each image category, we randomly selected five post-unlearning models. Subsequently, we generated 10,000 images using the diffusion model and used them to compute FID in comparison with the COCO-10k dataset Lin et al. (2014).

## 6.2 METRICS EFFECTIVENESS EVALUATION

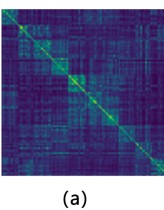 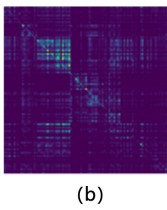 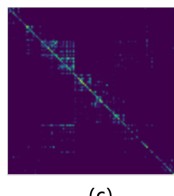

(a)            (b)            (c)

Figure 3: Visualization results of our metrics for 100 selected generated images and corresponding anchor images: (a) represents the Semantic Metric. (b) represents the Style Metric, and (c) represents the Total CPDM Metric. Brighter pixels indicate higher similarity between the two images, as determined by the formula $Softmax(-M^{100 \times 100})$. (c) demonstrates that the highlighted pixels are predominantly distributed near the diagonal line, effectively validating that our CPDM metric can comprehensively assess the similarity between two images. In addition, as ablation studies, (a) and (b) reveal that the Semantic metric is not sensitive to certain dissimilarities, while the Style metric is overly sensitive to some dissimilarities. The $w^{1 \cdots n}$ we choose is $[0.5, 0.1, 6e4, 4]$

After introducing our metric, it is crucial to validate its effectiveness. To validate our metric's effectiveness, we first visualize the results it produces. In this experiment, we selected 10 artists and generated images that mimic the style of their artworks, using the pipeline introduced in Section 4. We then randomly selected 10 generated images per artist, with visually similar features to their anchor image (potentially indicative of plagiarism), in total of 100 images. With these selections, we utilize our metrics to compute a matrix $M^{100*100}$, capturing the relationships between the 100 generated images and their corresponding 100 anchor images. The distributions of Semantic Metric, Style Metric and Total CPDM Metric are visualized in the Fig. 3. These visualizations evidently support the assertion that our metric can successfully identify images that may constitute plagiarism. Furthermore, it can also apply for the metrics in unlearning task, which allows for a more systematic and quantifiable assessment of the methods for unlearning.

## 6.3 RESULTS AND ANALYSIS

**Benchmark: Gradient Ascent-based Approach**

We have two primary foundational models in our repertoire. The first one is our finetuned model, SD-finetuned, which is employed to evaluate and assess the performance of the unlearning algorithm on the finetuned full parameters of the UNet within the stable diffusion framework. This evaluation includes both the model's ability to forget copyrighted images and its performance in generating other types of images. The second foundational model is an unlearning experiment conducted on the stable diffusion v2.1, which is currently the best publicly available generative model based on stable diffusion. During the unlearning experiments conducted on these two foundational models, for each image to be forgotten, the learning rate for gradient ascent is set at 5.0e-05, and unlearning training is performed for five epochs.

**Benchmark: Weight Pruning-based Approach**

Similar to gradient ascent, we conducted unlearning experiments on both foundational models using our dataset. For the SD-finetuned model, which has been fine-tuned on specific illustration styles, we

Table 2: Benchmark Performance Testing. For "Origin", we calculate the FID between the generated images from SD-v2.1 and the COCO-10K datasetLin et al. (2014). For the column "FID", we calculated the FID between the generated images and the COCO-10K dataset Lin et al. (2014). And "CPDM Metric" is measured between the anchored image and the generated images of corresponding models. Additionally, for the unlearning process, we uniformly measured the FID and CPDM Metric after applying the "Prune" and "Gradient Ascent" methods. The higher the CPDM Metric, the better the forgetting effect on copyright images. CM stands for CPDM Metric. Further detailed setups for ESD and Forget-Me-Not are provided in the supplementary.

| | Origin | | Prune | | Gradient Ascent | | ESD | | Forgot-Me-Not | |
|---|---|---|---|---|---|---|---|---|---|---|
| | FID↓ | CM↑ | FID↓ | CM↑ | FID↓ | CM↑ | FID↓ | CM↑ | FID↓ | CM↑ |
| Cartoon | 11.18 | 0.9110 | 11.34 | 26.0683 | 11.79 | 22.5369 | 15.72 | 2.0636 | 12.43 | 13.7545 |
| Portrait | 11.18 | 0.2153 | 11.49 | 4.2801 | 11.47 | 6.0105 | 18.24 | 2.2259 | 15.21 | 18.2984 |
| Wikiart | 11.18 | 0.1847 | 12.79 | 0.2689 | 12.86 | 14.1486 | 16.19 | 1.6044 | 12.42 | 16.1027 |

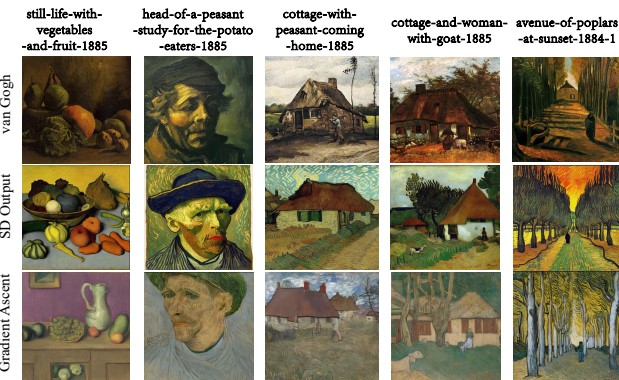

Figure 4: The unlearning experiment of Vincent van Gogh's art paintings on the benchmark.

set the pruning ratios $p_c\%$ and $p_w\%$ to 0.1 and 0.03, respectively. As for the SD-v2.1 model, we set the pruning ratios $p_c\%$ and $p_w\%$ to 0.1 and 0.005, respectively. The reason for employing a higher pruning ratio on SD-finetuned is due to the need for a more significant pruning ratio to forget such artistic styles when the model has been fine-tuned on a limited amount of data. For each image to be forgotten, we performed one epoch of iterative computation and pruning. During the pruning process, the optimizer remained disabled.

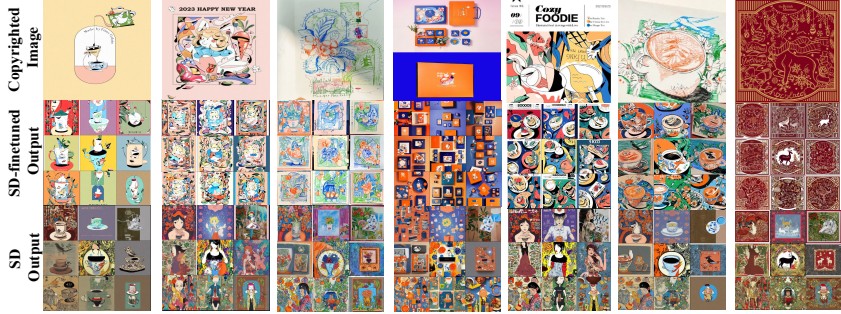

Figure 5: Illustration Images. Top row: Original copyrighted image. Middle row: Output from SD-finetuned. Bottom row: Output from SD-v2.1.

**Infringement Model: SD-Finetuned** By combining the prompts obtained from the pipeline described in Fig.2, we can simulate a model that utilizes stable diffusion to infringe upon specific artworks. Throughout the training phase of the SD-finetuned model, intended for generating infringing illustrations, we meticulously employed a set of 160 artwork images in conjunction with

Table 3: Metrics during the simulation process by illustration (Illu.) dataset. The "Origin" and "Finetuned" stand for the SD-V1.4 and the SD-finetuned on our Illustraion dataset. The other evaluation metrics are computed following the description in Table. 2. CM stands for CPDM Metric.

| | Origin | | Finetuned | | Prune | | Gradient Ascent | | ESD | | Forgot-Me-Not | |
|---|---|---|---|---|---|---|---|---|---|---|---|---|
| | FID↓ | CM↑ | FID↓ | CM↑ | FID↓ | CM↑ | FID↓ | CM↑ | FID↓ | CM↑ | FID↓ | CM↑ |
| Illu. | 13.21 | 1.0170 | 9.33 | 0.1534 | 36.93 | 2.0856 | 11.85 | 2.6615 | 8.61 | 2.0106 | 12.95 | 8.2107 |

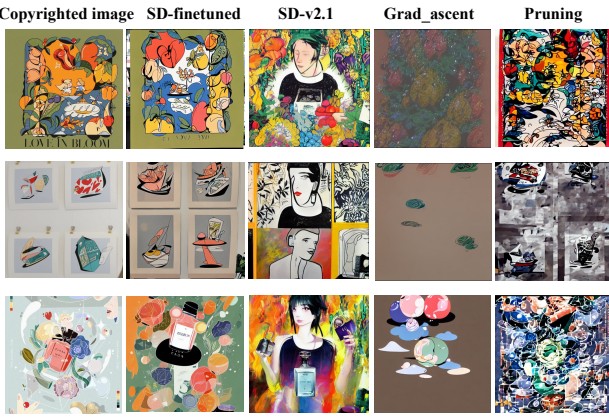

Figure 6: Experimental Results of Model Unlearning. Each row represents an illustration sample, where from left to right, it denotes the original image, the image generated after fine-tuning with Stable Diffusion, the image generated with SD-v2.1, and the image generated after unlearning using gradient ascent and pruning methods.

their corresponding prompts. To ensure a more faithful simulation of the process of infringing upon artwork, we opted for a two-stage fine-tuning procedure. This meticulous approach was implemented to yield infringing images that bear a striking resemblance to the original artworks in terms of their artistic characteristics, encompassing style, lines, lighting, composition, brushstrokes, and more. The specific training details are provided in the appendix. Additionally, there have been notable methods that attracted attention for using a small amount of data to fine-tune stable diffusion models, such as Lora Hu et al. (2021), DreamBooth Ruiz et al. (2023) and TextInversion Gal et al. (2022). These methods demonstrate remarkable generation results through minimal parameter updates, exhibiting excellent performance in terms of training efficiency and output quality. Nonetheless, given the distinctive nature of the artistic style employed by the artists (Anonymous Artist), characterized by uncommon elements, lines, and brushstrokes that have limited presence in the foundational stable diffusion training set, it appears that merely fine-tuning a subset of parameters might prove inadequate in generating infringing images. This development presents encouraging implications for copyright safeguarding. However, it remains evident that stable diffusion models inherently possess the capacity to assimilate any image style. Henceforth, we shall meticulously refine the complete spectrum of parameters within the Unet model to replicate the intricate nuances of infringement within the realm of artwork.

## 7 CONCLUSION

The remarkable generation and data fitting capabilities of large models like diffusion models have garnered significant attention. However, they have also raised concerns regarding image copyright and privacy. This work introduces a new large-scale dataset and benchmark focused on copyright protection, making it the first dataset in this domain based on diffusion models. Additionally, we provide a standardized metric for determining whether generated images infringe on copyrights. We hope that this dataset and benchmark will serve as a valuable resource and inspire new research directions in the field of copyright protection for artistic works.

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
