# A Dataset and Benchmark for Copyright Protection from Text-to-Image Diffusion Models

## A Appendix

### A.1 Scope and Contributions

**Scope:**

The scope of our paper focuses on:

- Frameworks for responsible dataset development. Our work focused on the frameworks for responsible dataset development to provide guidelines and principles to ensure that data collection methods are transparent, fair, and respectful of privacy.

- Audits of existing datasets. Audits of existing datasets also play a crucial role in ensuring copyright protection. These audits examine the origin and usage rights of the data within the dataset, verifying that the data has been obtained and used in compliance with copyright laws and regulations.

- Identifying significant problems with existing datasets and their use.

**Contributions:**

The contributions of our paper are summarized as follows:

- We introduce the CPDM dataset, which contains 21,000 images, with 2,100 anchor images and 18,900 generated images. It includes 1,500 images in the style category, 200 in the portrait category, 200 in the artistic creation figure category, and 200 in the licensed illustration category.

- The dataset is open-source and accessible to anyone, with long-term maintenance and updates. The link can be found at A.2.

- An open-source project website has been established, providing instructions on dataset usage and a comprehensive guide to the dataset generation process. The public is encouraged to participate and contribute new samples to the dataset on an individual basis. The project homepage can be found at *zhouq.net* and will gradually improve over time.

- The increasing global popularity of AIGC highlights the importance of privacy and copyright issues. AI companies such as OpenAI have responded to concerns around data security. At the national level, the US has proposed a new government agency in charge of approving large-scale AI models. Furthermore, the Chinese Cyberspace Administration has published a document emphasizing AIGC security issues. Therefore, the introduction of the CPDM dataset and benchmark, as the first dataset based on diffusion models, serves as a positive catalyst for the development of copyright protection in the AIGC era.

### A.2 Dataset hosting and maintenance

Public access and download links to the CPDM dataset are provided through the webpage: *http://149.104.22.83/unlearning.tar.gz*. It contains *.jpg* or *.png* files of all images and corresponding *Prompt* files, as well as generated infringing images. Publicly available code to provide reference code for using the dataset and computing the evaluation metrics will be released at *https://github.com/###*. The code repository additionally includes code to reproduce some of the methods evaluated in the paper. The CPDM dataset is hosted on the server.

### A.3 LICENSE

The images included in the CPDM dataset are either publicly available on the web or from three sources, Wikiart, Wikipedia, and Illustrator Anonymous Artist. The corresponding licenses for the ones that are available on the web are public domain, public domain, and illustrator <Anonymous Artist> , respectively. We do not own their copyrights. We, the authors of this paper and creators of the dataset, bear all responsibility in case of violation of rights.

### A.4 OUR ARTIST

Information is anonymous.

### A.5 CPDM METRIC DETAILS

After establishing the fundamental principle of considering metrics from both semantic and stylistic perspectives, the factors determining the final form of metrics are reduced to two parts: the weight of the gram matrix loss in each layer of the style metric $\{w^l\}$ and the formula for calculating the final metric based on semantic and style metric.

For the first part, our approach is to utilize weights to approximate the normalization of loss between layers, ultimately enabling the style metric to achieve the desired effect on the selected similar paired data, that is, higher similarity for paired works and higher similarity for works by the same author.

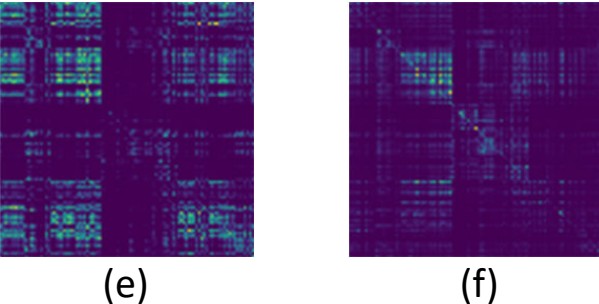

(e)                              (f)

Figure 1: (a) $\{w^l\} = [1., 1., 1., 1.]$. (b) $\{w^l\} = [0.5, 0.1, 6e4, 4]$ The two figures visualize the distribution of style loss, other visualization settings are the same as Fig.3 in the paper. We can observe that， by adopting the selected weights (b), the visualization results demonstrate superior performance. Specifically, the brightness of the pixels near the diagonal line is significantly higher, and there is a tendency to form bright clusters according to different authors.

For selecting generation formula, we test multiple different formulas to obtain the most significant effect. As can be seen from 2, in fact, various formulas have certain effects, and the effect of formula (c) is the most significant and is less likely to be greatly influenced by either part of the two parts ( The practical implication is that if either semantics or style is much too similar, it may be judged as plagiarism or infringement). Therefore, we choose formula (c) to calculate our final CPDM metric.

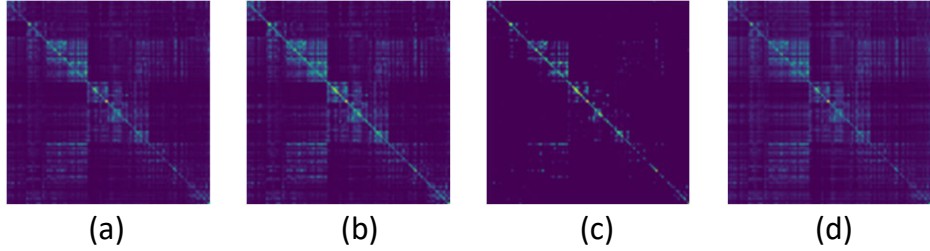

Figure 2: (a) $= (Loss_{sem} + Loss_{style})$, (b) $= (Loss_{sem}^2 + Loss_{style})$, (c) $= (Loss_{sem} * Loss_{style})^2$, (d) $= (Loss_{sem} + Loss_{style})$ The visualization settings are the same as Fig.3 in the paper.

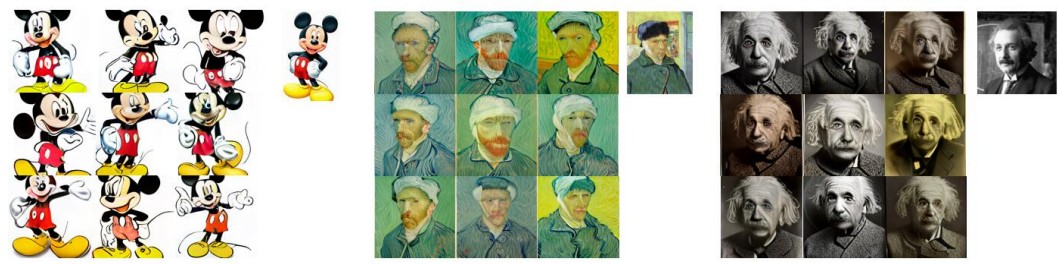

Figure 3: The experimental results of DALLE-mini are shown in the figure, with 9 generated images on the left and anchor images on the right. The corresponding CPDM metric data is presented in Table 1.

Table 1: The images generated using DALLE-mini are evaluated with the CPDM metric. The rows and columns of the table represent anchor images and generated images, respectively. A lower CPDM metric indicates a higher correlation between images. It can be observed that the images in the diagonal positions exhibit the highest correlation.

| CPDM Metric | Style | Portrait | Artistic Creation Figure |
|---|---|---|---|
| Style_g | **0.9930** | 22.4001 | 9.95463 |
| Portrait_g | 4.1052 | **0.9067** | 5.8540 |
| Artistic_Creation_Figure_g | 12.1819 | 21.8751 | **0.9170** |

## A.6   IMPLEMENTATION DETAILS

**Fine-tuning the Stable Diffustion based on Illustraions** Digital illustration is an important genre in modern art, widely used in advertising and promotion, website and mobile application design, digital publications, and other scenarios. Due to its characteristics such as digitization, diverse creativity, and rich and variable elements, digital illustration is easier to access and prone to being plagiarized.

Moreover, this methodology aims to exemplify the capability of models such as stable diffusion to replicate both the artistic style and content of artworks, even when provided with a limited collection of artists' works. In the first stage, we trained the model on stable diffusion v1.4 with a learning rate of 5.0e-05 for 30 epochs. The training dataset consisted of a subset (10,000 images) from the laion 5B Schuhmann et al. (2022) dataset mixed with the 160 illustration images. The other parameters were set to the default values provided by stable diffusion repository. In this stage, our goal was for the model to learn the artistic style and elements while preserving the diversity of its generation's capabilities. In the second stage, we fine-tuned the model on the 160 illustration dataset for 6 epochs using a learning rate of 1.0e-05. This resulted in a stable diffusion model (SD-finetuned) that has the ability to generate works extremely similar to those of the artists.

Finetuning the parameters of the UNet module in the stable diffusion, it is effortless to achieve plagiarism of specific artistic styles. The generated images from the model are almost indistinguishable from the original copyrighted images in terms of artistic style, elements, and composition. This demonstrates the substantial impact of large models on copyright infringement in artistic creations.

**Utilization of ChatGPT**

The primary application of ChatGPT was during the collection of anchor image data, where it facilitated the creation of a list containing the most renowned and potentially infringing images. This approach streamlined the process of efficiently gathering the most probable anchor images. Furthermore, we experimented with employing ChatGPT to refine prompts generated by the Clip Interrogator. Leveraging the advanced textual processing capabilities of large language models, we aimed to optimize prompts by addressing issues such as semantic ambiguity, lack of coherence, and incomplete phrasing. The goal was to produce prompts that exhibited a closer alignment with the input image. Regrettably, the experimental outcomes demonstrated that the integration of ChatGPT-optimized prompts did not result in substantial improvements. Specifically, the enhanced prompts failed to significantly aid in identifying more closely related infringing images to the input image. Instead, this integration led to an escalation in pipeline complexity and necessitated consideration of response time and stability concerns associated with the large language model. Consequently, in the final version of our work, we restricted the invocation of ChatGPT solely to the process of collecting anchor images.

**Data Collection Method and Data Quality**

During the process of data collection, we employed a combined approach involving manual validation, ChatGPT, and CPDM metric to ensure the quality of the collected data. Specifically, we initiated the process by utilizing ChatGPT to compile a list of the most renowned and likely infringing image names. This list was curated to encompass a diverse range of image types. Subsequently, a manual review was undertaken to eliminate anchor images lacking in representativeness. For images generated by the model, we employed a dual strategy involving the CPDM metric and manual assessment. This combination served to guarantee the accuracy and representativeness of the collected images. Moreover, it is crucial to recognize that the method of data collection has a direct impact on the overall quality of the dataset. The CPDM metric played a crucial role as an initial screening tool, while the subsequent manual review ensured the ultimate quality of the dataset. The synthesis of these approaches not only ensured the data's integrity but also contributed to a dataset that better reflects the diversity and accuracy required for the research.

**Baseline for Unlearning Algorithm Design**

We have proposed two fundamental unlearning methods for text-to-image generation models: the Gradient Ascent-based Approach and the Weight Pruning-based Approach. During the unlearning experiments, we have kept the parameters of the text embedding module and the autoencoder module frozen, focusing solely on adjusting the model's Unet structure. Taking the example of stable diffusion, the model's text embedding module employs either "openai/clip-vit-large-patch14" or "ViT-H-14," both of which are pretrained modules. Consequently, during the fine-tuning of the diffusion model, it's common practice to freeze these parameters to maintain the semantic information of the text module and the feature information of the image encoder-decoder module. Similarly, in conducting unlearning experiments with copyrighted images, based on insights from the esd [reference] and forgot-me-not [reference] papers, we've found that the cross-attention structure of the Unet module greatly influences the connection between prompts and the semantic and stylistic information of generated images. Adjusting these parameters through unlearning enables the model to forget copyrighted images without excessively compromising its generative capability. Thus, we've introduced two fundamental unlearning methods specifically targeting the latent space Unet module.

**Forgetting Copyrighted Images**

For text-to-image generation models, when we identify that the model has generated copyrighted images based on a specific prompt and we intend to forget those corresponding copyrighted images, a two-step process is employed. Firstly, the relevant unlearning algorithm is applied to the generation model. Subsequently, the unlearned model is employed with the same prompt to regenerate the corresponding image. Following this, the CPDM metric is utilized to measure the similarity between

the generated images before and after the unlearning process. This assessment helps determine whether the model has successfully forgotten the corresponding copyrighted images.

**Experimental Setups for ESD and Forget-Me-Not**

In the evaluation of both unlearning algorithms, the open-source code for ESD is tailored for Stable Diffusion v1.4. For exploratory experiments, we made appropriate code adjustments to accommodate Stable Diffusion v2.1, while maintaining other parameter configurations consistent with those specified in Table 2 of the paper. We utilized 100 iterations for ESD due to observed limitations in image quality and excessive decline in FID when using the official 1000 iterations. This could be attributed to the disparities in model versions between Stable Diffusion v1 and Stable Diffusion v2. Regarding the Forget-Me-Not experiment, we introduced modifications to the official parameter settings. The default image size for the official experiment is 512, employing the "stable-diffusion-2-1-base" model. We adapted this to an image size of 768 and employed the "v2-1-768-ema-pruned" model to ensure consistency with the baseline methods used in several experiments outlined in Table 2 of the paper. It's worth noting that, in both experiments, the code and parameter adjustments were not meticulously fine-tuned, potentially affecting the representativeness of the experimental outcomes for the optimal efficacy of the proposed approach.

## A.7 RELEVANT WORK IN ARTISTIC IMAGE COMMUNITIES

There has been considerable attention towards community efforts that employ Stable Diffusion models to imitate artistic styles. Notable examples include websites like https://stablediffusion.fr/artists and https://www.urania.ai/top-sd-artists. These platforms curate collections of images that bear stylistic resemblance to the works of over a thousand artists, spanning both contemporary and classical art. In the workflow of these image communities, artistic style images are generated by manually providing a prompt, where the artist's name is incorporated as a style cue in the prompt, such as "A woman + [artist]." This approach emphasizes capturing the unique artistic traits of an artist's style. In our framework, our approach begins by utilizing existing, specific, and valuable artistic images as anchor images. Unlike arbitrarily determining a subject or content for a painting, we derive corresponding prompts through multimodal analysis of these anchor images. These prompts are then combined with the artist's name and fed into a text-to-image model to generate images that are both content-wise and stylistically similar to the anchor image. For instance, we might create a prompt like "fritillaries-in-a-copper-vase-1887, by vincent-van-gogh, painting of a vase with fritillaria flowers in it, in the starry night, orange flowers, pine, luscious brushstrokes, prizewinning, cone, high details, hips, tyler, mesmerizing, description, pot, visually stunning, unlit." This prompt includes a description of the painting's content, its title, and the artist's name, enabling the identification of corresponding infringing images. In essence, our method involves collecting authentic, valuable, and specific images from the art world to be used as training examples for image generation models. This represents a more rigorous form of style imitation. In contrast, the artistic images in the provided links tend to focus on capturing certain aspects of an artist's style, falling into the category of broader style imitation. Furthermore, certain images from these links closely resemble those in our dataset, such as the character image generated from the prompt "A woman, [Vincent Van Gogh]."

## A.8 IMAGE METRICS AND HUMAN PERCEPTION ALIGNMENT

We conducted the following experiment to validate the correspondence between the CPDM metric and human perceptual evaluation. Specifically, we randomly selected 10 anchor images from each category in the CPDM dataset, resulting in a specialized dataset of 40 images for this experiment. We then divided the prompts corresponding to each image into three different lengths: short, medium, and long. These prompt lengths capture different levels of completeness in describing the anchor images. Using these prompts, we generated corresponding counterfeit images and obtained metric values. In the final step, we had human evaluators manually rate the similarity between the images generated from prompts of varying lengths and the corresponding anchor images. We then compared these human perceptual ratings with the metric values and conducted a correlation analysis. The experimental results demonstrated a significant alignment between the benchmark's metric values and human assessments of image counterfeiting trends. Experimental results refer to Table 2 and Figure 6.

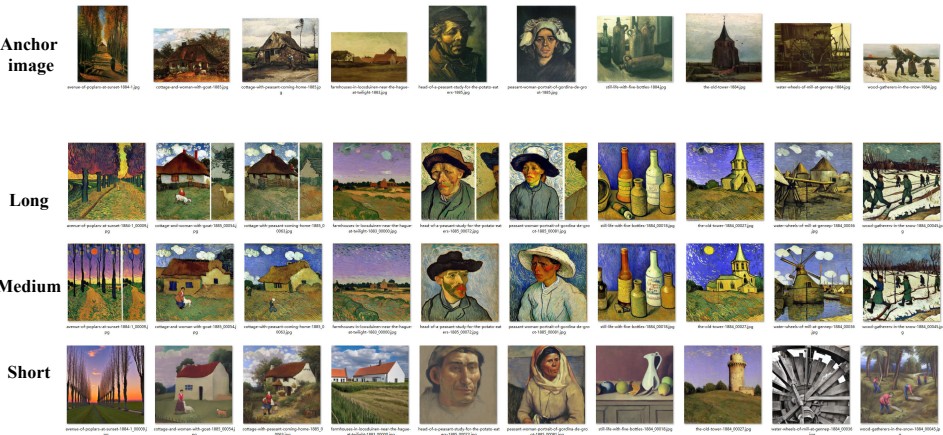

Figure 4: Style: figures generated under prompts of different lengths.

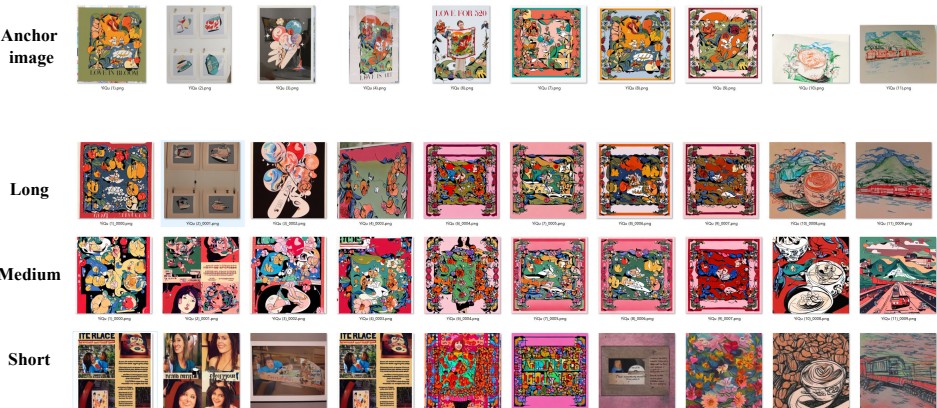

Figure 5: Illustration: figures generated under prompts of different lengths.

## A.9 BALANCING AMONG DIFFERENT USERS

Balancing the interests between users of art and imitators with the original creators of art is a practical challenge. For example, imitators argue that their derivative works incorporate many of their own design elements, making them significantly distinct from the original image. On the other hand, original creators believe that imitated images still contain their own creativity and artistic achievements. The CPDM dataset currently encompasses four major categories of images: art paintings, cartoons, portraits, and illustrations. Our goal is to protect a broader range of image types. Through the demo, we aim to establish a community that can accommodate various image categories and styles, facilitated by extensive voluntary searches by users and artists. This necessitates the provision of evaluation metrics to determine whether infringement has occurred. These metrics should be reliable, consistent, widely applicable to diverse image categories and styles, and generally accepted by the majority. For instance, an image that has been legally determined to be infringing due to copyright disputes might not be perceived as such by its imitator, who believes their imitation includes their knowledge and creativity. However, according to prevailing universal standards (which could be legal), the imitation indeed constitutes infringement. Regarding the category of art paintings, artists often consider inspiration and creativity as the most valuable aspects of a piece, while the style, lines, and other elements are expressions of their thoughts, techniques, and habits. Imitators, through simple modifications or style transfers, might perceive the modified image to be significantly different from the original, leading them to believe it doesn't infringe copyright. However, by widely accepted standards, the modified image might still be deemed an infringement. Therefore, we aim to provide specific evaluation metrics rather than ambiguous or multiple indicators. Despite the fact that

Table 2: Correlation Analysis of mage Metrics and Human Perception. The table lists the specific CPDM metric values, corresponding to Figure 6.

**Licensed Illustration**

| Sample | 1 | 2 | 3 | 4 | 5 | 6 | 7 | 8 | 9 | 10 |
|--------|------|------|------|------|------|------|------|------|------|------|
| Long | 0.0199 | 0.4043 | 0.3248 | 1.3084 | 0.3662 | 0.3272 | 0.0170 | 0.9741 | 0.3277 | 0.1064 |
| Midium | 0.1424 | 6.1508 | 5.6219 | 3.6961 | 11.1213 | 23.4006 | 0.0607 | 1.3994 | 14.0937 | 0.0803 |
| Short | 2.1791 | 1.4756 | 5.0242 | 20.7223 | 1.6771 | 48.3086 | 33.6176 | 3.5753 | 1.3758 | 2.5866 |

**Portrait**

| Sample | 1 | 2 | 3 | 4 | 5 | 6 | 7 | 8 | 9 | 10 |
|--------|------|------|------|------|------|------|------|------|------|------|
| Long | 15.4628 | 0.6014 | 0.6022 | 1.2449 | 1.2597 | 0.6904 | 3.0976 | 1.4410 | 15.5803 | 1.1136 |
| Midium | 7.9956 | 0.8865 | 0.2389 | 1.9183 | 1.8830 | 5.4410 | 4.8874 | 3.1025 | 19.2009 | 7.8166 |
| Short | 9.0726 | 0.9197 | 0.9680 | 1.5752 | 1.4851 | 0.6818 | 4.2184 | 1.4081 | 8.3657 | 2.0543 |

**Style**

| Sample | 1 | 2 | 3 | 4 | 5 | 6 | 7 | 8 | 9 | 10 |
|--------|------|------|------|------|------|------|------|------|------|------|
| Long | 2.0524 | 0.9340 | 3.7435 | 1.3003 | 5.1592 | 1.3929 | 1.8341 | 1.5736 | 1.4996 | 1.4300 |
| Midium | 5.1566 | 1.7213 | 2.8873 | 1.6611 | 17.9385 | 1.3685 | 1.0523 | 2.4432 | 1.5736 | 0.8473 |
| Short | 11.7106 | 2.2211 | 0.4889 | 3.4210 | 6.0821 | 0.4062 | 5.6952 | 5.5935 | 0.8287 | 8.4373 |

**Artistic Creation Figure**

| Sample | 1 | 2 | 3 | 4 | 5 | 6 | 7 | 8 | 9 | 10 |
|--------|------|------|------|------|------|------|------|------|------|------|
| Long | 3.8000 | 0.3160 | 100.0000 | 0.2873 | 8.7474 | 1.1633 | 4.5712 | 0.4855 | 0.6338 | 3.7595 |
| Midium | 0.7875 | 2.6698 | 6.9022 | 0.9813 | 8.6100 | 29.8275 | 6.4484 | 0.7571 | 3.4068 | 3.6385 |
| Short | 68.3414 | 12.4492 | 76.5174 | 61.7901 | 10.2511 | 5.9912 | 3.1749 | 4.2498 | 4.8176 | 5.4505 |

copyright disputes involve varying perspectives from creators and imitators and have some gray areas, we aspire to propose metrics that align as closely as possible with universally accepted evaluation criteria. We maintain close discussions with the artists within our team, continuously improving and refining our evaluation methods. Our aim is for the outcomes of our evaluation criteria to closely align with the assessments made by artists.

## A.10 USAGE OF COPYRIGHTED IMAGES

The current dataset includes copyrighted content obtained from Wikipedia. The images provided by this community have shared copyrights and can be accessed and utilized for non-commercial purposes and educational research (https://www.wikiart.org/en/terms-of-use). Additionally, a portion of illustrative images comes from the Anonymous Artist . These illustrative images are also available for non-commercial and educational research purposes. For commercial usage, please reach out to Anonymous Artist at https://###/ to request authorization. After the demo link is launched, the dataset will gradually expand to encompass various other categories of images. Through the engagement and usage of a wide range of users, we will curate and collect images that may have been subject to copyright infringement. The collection of such images is carried out with the authorization of the image providers, adhering to non-commercial use and scientific research purposes.

## A.11 PROSPECTS FOR DATASET SCALE AND DIVERSITY

Currently, our research focus is restricted to the domain of two-dimensional image copyright to analyze and explore the infringement scenarios where copyright images are used in training datasets for text-to-image generation models. This is a highly relevant issue that has attracted significant attention. The data collection and preparation required for the development of large models have a profound impact on image copyright issues. For instance, training sets for image generation models like Stable Diffusion include massive amounts of natural images, some of which unavoidably include copyrighted content. As a result, issues related to copyright infringement in the context of two-dimensional content and text-to-image generation models have gained widespread attention recently. We acknowledge that real-world copyright infringement encompasses a broader spectrum, including three-dimensional content and multimedia presentations. However, attempting to cover all potential infringement domains is a complex and challenging endeavor. We have chosen to focus on the copyright issues of two-dimensional image content and text-to-image generation models, with

| Prompt Lentth | Sample | CPDM Metric | | | | Human Perception | | |
|---|---|---|---|---|---|---|---|---|
| | | Long | Midium | Short | | Long | Midium | Short |
| Licensed Illustration | 1 | 1 | 2 | 3 | | 1 | 2 | 3 |
| | 2 | 1 | 3 | 2 | | 1 | 2 | 3 |
| | 3 | 1 | 3 | 2 | | 1 | 2 | 3 |
| | 4 | 1 | 2 | 3 | | 1 | 2 | 3 |
| | 5 | 1 | 3 | 2 | | 1 | 2 | 3 |
| | 6 | 1 | 2 | 3 | | 1 | 2 | 3 |
| | 7 | 1 | 2 | 3 | | 1 | 2 | 3 |
| | 8 | 1 | 2 | 3 | | 1 | 2 | 3 |
| | 9 | 1 | 3 | 2 | | 1 | 2 | 3 |
| | 10 | 1 | 2 | 3 | | 1 | 3 | 2 |
| Portrait | 11 | 2 | 3 | 1 | | 2 | 1 | 3 |
| | 12 | 1 | 2 | 3 | | 1 | 2 | 3 |
| | 13 | 2 | 2 | 4 | | 3 | 1 | 2 |
| | 14 | 1 | 3 | 2 | | 1 | 3 | 2 |
| | 15 | 1 | 3 | 2 | | 1 | 3 | 2 |
| | 16 | 1 | 3 | 2 | | 1 | 2 | 3 |
| | 17 | 1 | 3 | 2 | | 1 | 2 | 3 |
| | 18 | 1 | 3 | 2 | | 1 | 2 | 3 |
| | 19 | 3 | 1 | 2 | | 1 | 2 | 3 |
| | 20 | 1 | 3 | 2 | | 1 | 2 | 3 |
| Style | 21 | 1 | 2 | 3 | | 1 | 2 | 2 |
| | 22 | 1 | 2 | 3 | | 1 | 2 | 3 |
| | 23 | 3 | 2 | 1 | | 1 | 2 | 3 |
| | 24 | 1 | 2 | 3 | | 1 | 2 | 3 |
| | 25 | 1 | 3 | 2 | | 1 | 2 | 3 |
| | 26 | 3 | 1 | 2 | | 1 | 2 | 3 |
| | 27 | 2 | 1 | 3 | | 1 | 2 | 3 |
| | 28 | 1 | 2 | 3 | | 1 | 2 | 3 |
| | 29 | 3 | 1 | 2 | | 2 | 1 | 3 |
| | 30 | 2 | 1 | 3 | | 1 | 2 | 3 |
| Artistic Creation Figure | 31 | 2 | 1 | 3 | | 1 | 2 | 3 |
| | 32 | 1 | 2 | 3 | | 1 | 2 | 3 |
| | 33 | 2 | 3 | 1 | | 1 | 2 | 3 |
| | 34 | 1 | 2 | 3 | | 1 | 2 | 3 |
| | 35 | 1 | 2 | 3 | | 1 | 2 | 3 |
| | 36 | 1 | 3 | 2 | | 3 | 1 | 2 |
| | 37 | 3 | 2 | 1 | | 1 | 2 | 3 |
| | 38 | 1 | 2 | 3 | | 1 | 2 | 3 |
| | 39 | 1 | 2 | 3 | | 1 | 2 | 3 |
| | 40 | 1 | 2 | 3 | | 1 | 2 | 3 |

Figure 6: Correlation Analysis of mage Metrics and Human Perception. As the color deepens, the correlation increases. This observation highlights the alignment between the trend of the CPDM metric and human perception

the aim of laying the groundwork for future expansion into other dimensions of image infringement. We hope that future work can expand the scope to encompass three-dimensional and multimedia content, thus conducting a comprehensive analysis of copyright infringement issues across various dimensions. Meanwhile, the demo for our proposed method will be launched soon and continuously improved. With the demo link, we hope that any artist or individual can upload images they believe to be infringing. If users are willing to share their images with the dataset, we will add infringing images to the CPDM dataset through a dual process of evaluation metrics and human screening. We believe that the dataset's scale will become richer as more people participate, and we will rigorously control the dataset's quality and make timely revisions. Once again, we sincerely appreciate your constructive feedback.

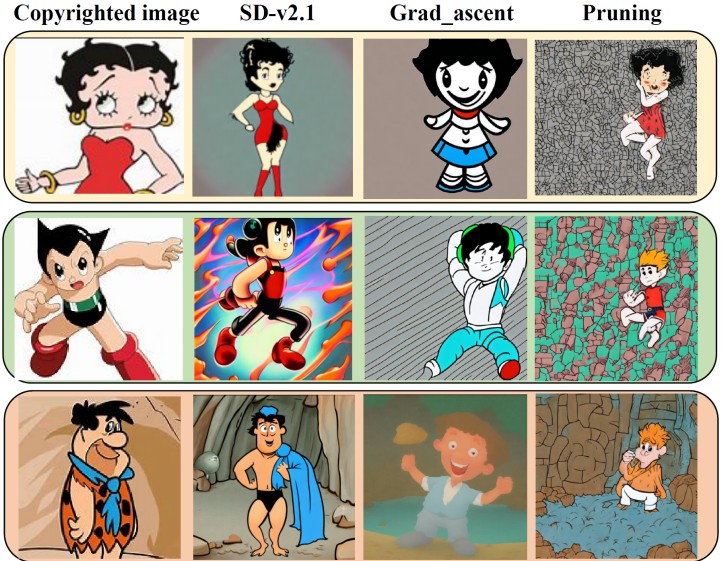

Figure 7: benchmark experiments on the *Artistic Creation Figure*.

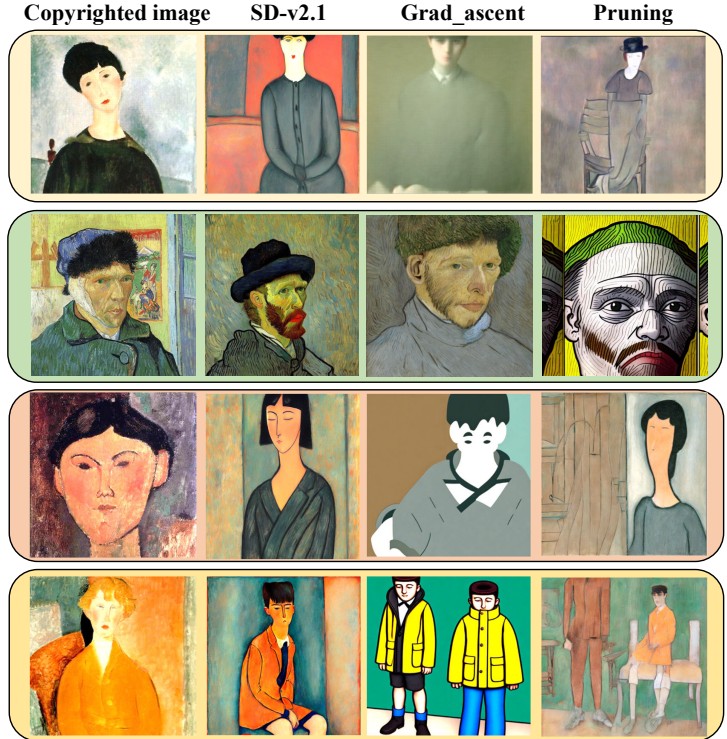

Figure 8: Benchmark experiments on the *WikiArt*.

## A.12   ADDITIONAL RESULTS OF BENCHMARK

## A.13   LIMITATIONS AND NEGATIVE SOCIETAL IMPACTS

- The scale of the dataset is relatively limited, which may fail to adequately cover a wide range of copyright images and associated cues. This could result in suboptimal performance of the forgetting algorithm when dealing with copyright images not included in the dataset.

| Copyrighted Image | SD-v1.4 | Before Unlearning
*SD-finetuned* | After Unlearning
*Erase* |
|---|---|---|---|

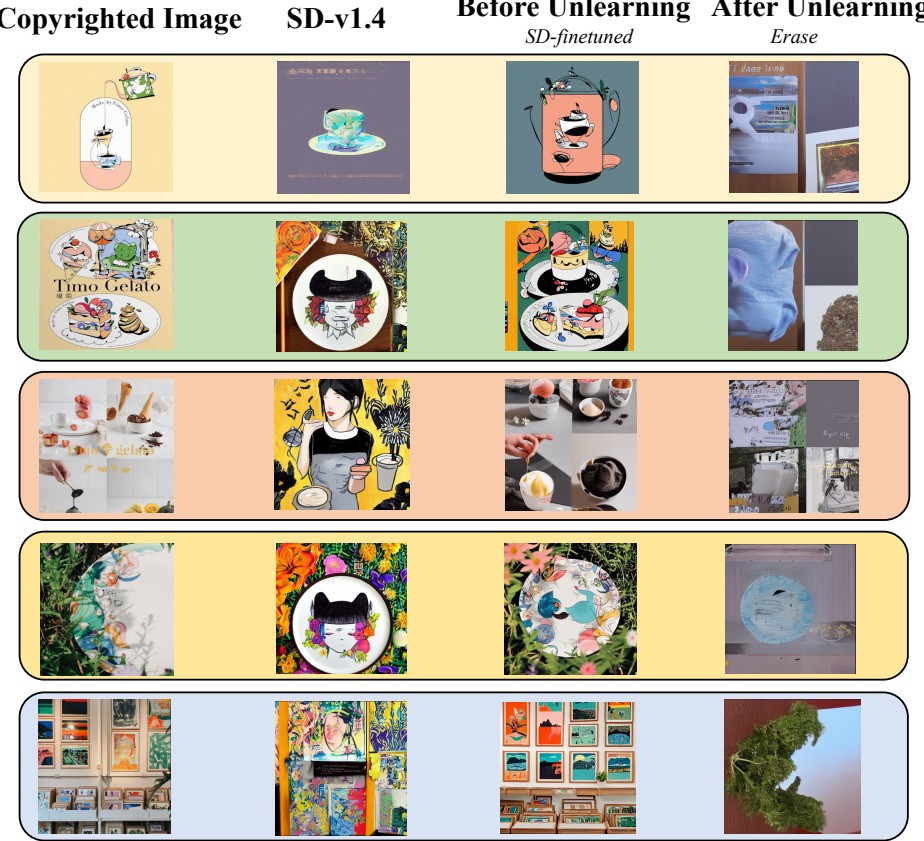

Figure 9: Benchmark experiments with unlearning method EraseGandikota et al. (2023).

Table 3: The prompt for the unlearning experiment of Vincent Van Gogh's art painting in Fig. 4.

| | |
|---|---|
| | *"still-life-with-vegetables-and-fruit-1885, by vincent-van-gogh, painting of a ... vegetables, 19 century, 3 heads, early 1 9 century, salad, right side composition, chalk, corner"* |
| | *"head-of-a-peasant-study-for-the-potato-eaters-1885, ... open, tilted head, vert coherent, february), farmer, yellowed with age, grain"* |
| **Prompts**

**<Van Gogh>** | *"cottage-with-peasant-coming-home-1885, by vincent-van-gogh, painting of a man standing in front of a thatched ... milkman, a brick cabin in the woods, poor buildings, pastelle, wandering"* |
| | *"cottage-and-woman-with-goat-1885, by vincent-van-gogh, a painting of a woman and a child outside a thatched ... detail, 1852, tonalist style, details"* |
| | *"avenue-of-poplars-at-sunset-1884-1, by vincent-van-gogh, a painting of a person walking down a path in the woods, by Van Gogh, church, at sunset in autumn, driveway, green ... siècle, pot, amsterdam, definition"* |

To enhance the algorithm's accuracy and generalizability, it may be necessary to expand the dataset and incorporate more diverse images and cues.

- The quality of the dataset is paramount to the effectiveness of the forgetting algorithm. If the dataset contains erroneous or inaccurate matches, it can hinder the algorithm's ability to correctly identify and handle copyright images. Therefore, ensuring the quality and accuracy of the dataset is of utmost importance.

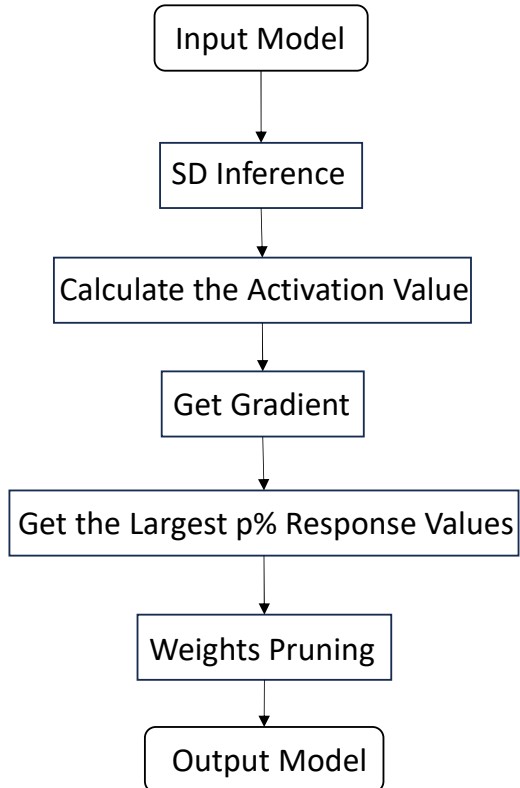

Figure 10: Activation-based pruning method.

- Dataset bias can impact the fairness and accuracy of the forgetting algorithm if there are evident biases in the copyright images and cues within the dataset. To mitigate this issue, it is crucial to ensure that the dataset is broadly representative and diverse.

- Legal and ethical considerations arise when employing copyright images as part of the dataset. When collecting and utilizing such images, it is essential to comply with relevant regulations and respect the rights of the original creators.

- Due to CLIP's inherent limitations including dataset biases, incomplete semantic and texture comprehension, and weak expression of certain image features, it fails to capture all image characteristics and fine details during the image-to-text process. The accuracy of prompts constrains the generation model's ability to produce copyrighted images.

- The use of generic CLIP and Inception as the primary feature extractors for image similarity metrics is subject to limitations stemming from the restricted quantity and variety of pretraining data available to these extractors. Consequently, inherent dataset biases and feature extraction tendencies are present, leading to an inability to capture certain unique or previously unseen image features. This situation may potentially restrict the applicability of evaluation metrics on specific style images.

## A.14 RESOURCE CONSUMPTION

**Human Resources:** A team of 13 members, including both computer scientists and artists.

**Computational Resources:** (4 * 24 * 30) hours (Nvidia A100).

Table 4: The prompts for illustration image in Fig. 5.

| | |
|---|---|
| **Prompts** | *"Illustrator <Anonymous Artist> , illustrations,there is a cartoon of a bird that is ... cover, uncropped, cut-away, watering can, minimal composition, by Masolino, loosely cropped, label"* |
| | *"Illustrator <Anonymous Artist> , illustrations,there is a cartoon picture of a woman with a ... colored accurately, year 2023, a blond, very very happy!, illustratioin,中元节"* |
| | *"Illustrator <Anonymous Artist> , illustrations,someone is holding a drawing of a flower in a ... al fresco, hard morning light, panoramic shot, fuchsia and blue, stipple, 2021, not blurry"* |
| **<Anonymous Artist>** | *""Illustrator <Anonymous Artist> , illustrations,there are a lot of different items that are on ... atmosphere, box, bautiful, star born, michelin restaurant, vivid)"* |
| | *"Illustrator <Anonymous Artist> , illustrations,a close up of a magazine cover with a woman in a dress, amazing ... Bowler, bt21, idyllic, eating, ffffound, inspired by Olive Mudie-Cooke, tummy"* |
| | *"Illustrator <Anonymous Artist> , illustrations,there is a drawing of a cup of coffee with a bird ... promo image, 1956, delightful surroundings, doodles"* |
| | *"Illustrator <Anonymous Artist> , illustrations,a red and gold christmas card with a horse and other holiday ... poster, an engraving, 2019, chocolate, epicurious, cd, album, sk, hello, english, tablecloth, b"* |

Table 5: The exemplification of illustration prompts. The prompts listed in ascending order within the table align harmoniously with the images sequentially depicted in Fig. 6.

| | |
|---|---|
| **Prompts** | *"Illustrator <Anonymous Artist> , poster of fruits, created by James Jean and Victo Ngai. Represents love as the beginning of all. ,..., Tom Whalen's blooming style in 8K resolution."* |
| | *"Illustrator <Anonymous Artist> , presents 4 Jetsons-inspired illustrations of a woman, printed on paper and arranged on a wall. ,..., The scenes include icebergs, a carousel, and ships, screen printed as part of an 8-piece portfolio."* |
| **<Anonymous Artist>** | *"Illustrator <Anonymous Artist> , presents an illustration featuring a person holding balloons, a collaborative work by Tristan Eaton and Greg Rutkowski. ,..., It portrays a slice of life, capturing the essence of caramel."* |
| | *" Illustrator <Anonymous Artist> presents a poster featuring a woman holding a perfume bottle. ,..., Enclosed in a glass cover, the artwork captures a flower explosion in a snapshot with detailed shots. "* |

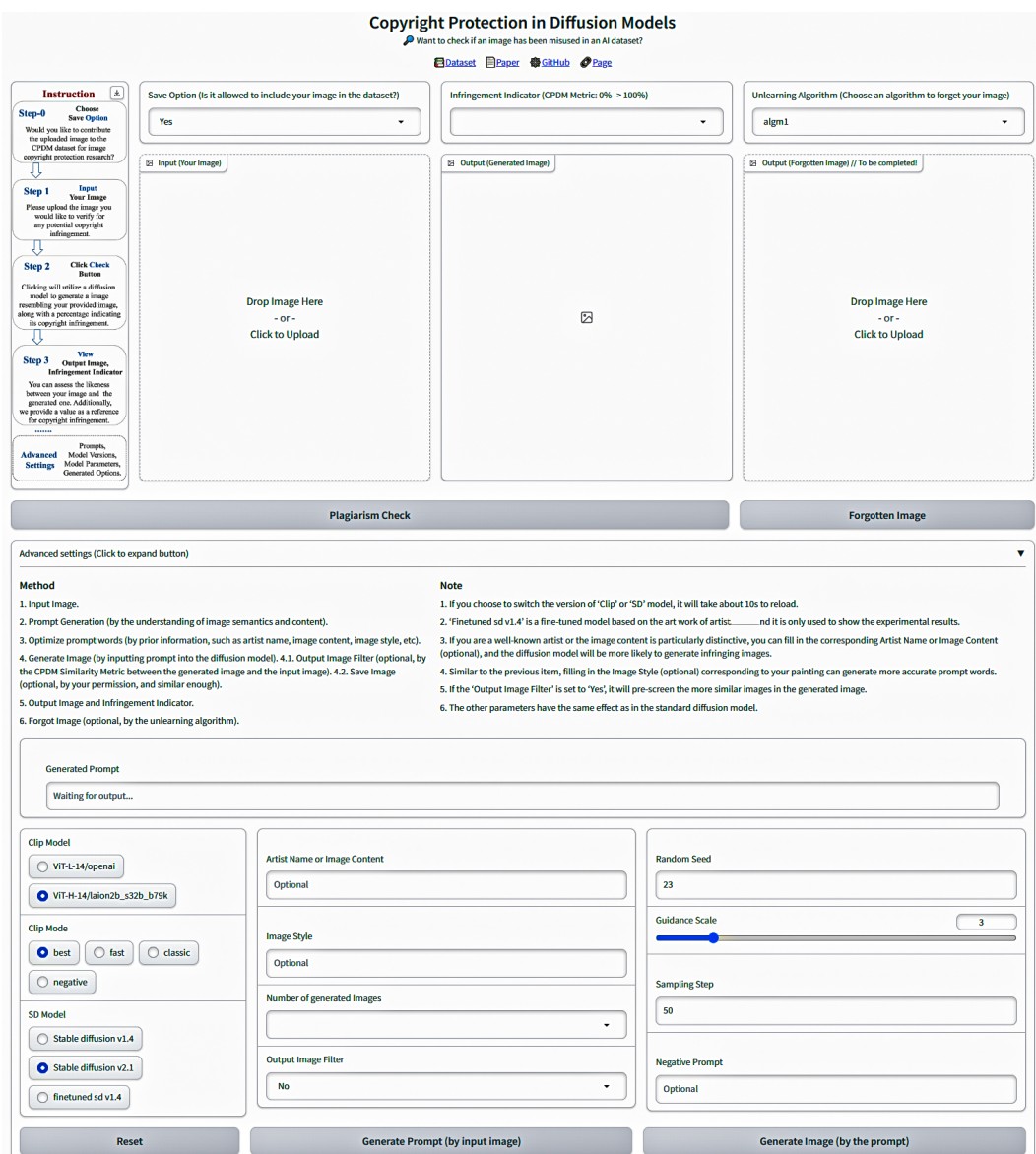

Figure 11: An example of our demo interface. The various functionalities will be continuously improved and updated.

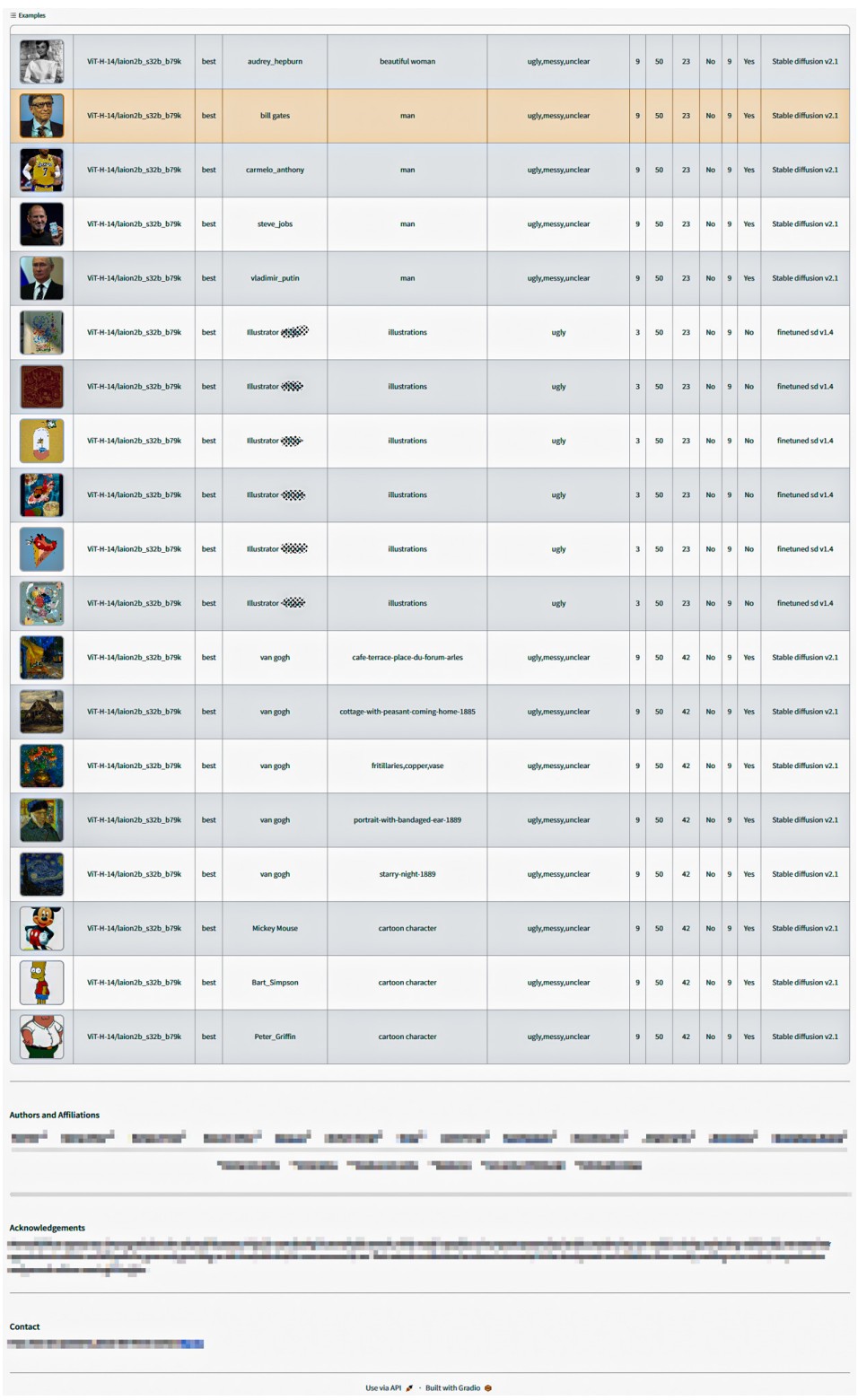

Figure 12: An example of our demo interface. The various functionalities will be continuously improved and updated.

CHECKLIST


## B  DATASHEETS FOR DATASETS

| **Motivation** |
|---|

**For what purpose was the dataset created?** Was there a specific task in mind? Was there a specific gap that needed to be filled? Please provide a description.

The dataset was created with the purpose of advancing research and development in the field of text-to-image generative models. These models aim to generate realistic images based on textual descriptions, effectively bridging the gap between language and visual content. However, the rapid advancements in text-to-image generation techniques have also raised concerns regarding copyright protection, such as the unauthorized learning of content, artistic creations, and portrait. We aim to develop a dataset and metrics that facilitate the identification of copyright infringement, while enabling a fair comparison of methods for mitigating such infringements.

**Who created this dataset (e.g., which team, research group) and on behalf of which entity (e.g., company, institution, organization)?**

This dataset was collaboratively created by researchers from Peking University, Tsinghua University, and University of California, Berkeley (UCB), as well as researchers from the industry, specifically ByteDance company.

**Who funded the creation of the dataset?** If there is an associated grant, please provide the name of the grantor and the grant name and number.

No.

**Any other comments?**

| **Composition** |
|---|

**What do the instances that comprise the dataset represent (e.g., documents, photos, people, countries)?** Are there multiple types of instances (e.g., movies, users, and ratings; people and interactions between them; nodes and edges)? Please provide a description.

The dataset primarily comprises anchor images, generated images and their corresponding prompts. The anchor images are initially collected as a set of images that potentially contain copyrighted content. These anchor images are then processed using the CLIP-interrogator, which yields prompts associated with each anchor image. Subsequently, the obtained prompts are utilized as inputs for the stable diffusion model, enabling the generation of images by the stable diffusion model.

**How many instances are there in total (of each type, if appropriate)?**

The dataset consists of 2100 anchor images, each accompanied by a corresponding prompt, resulting in a total of 2100 prompts. Using these prompts as input, a total of 18900 images were generated.

**Does the dataset contain all possible instances or is it a sample (not necessarily random) of instances from a larger set?** If the dataset is a sample, then what is the larger set? Is the sample representative of the larger set (e.g., geographic coverage)? If so, please describe how this representativeness was validated/verified. If it is not representative of the larger set, please describe why not (e.g., to cover a more diverse range of instances, because instances were withheld or unavailable).

The dataset encompasses the entirety of all possible instances.

**What data does each instance consist of? "Raw" data (e.g., unprocessed text or images) or features?** In either case, please provide a description.

Each instance within the dataset comprises an anchor image, a prompt, and nine corresponding generated images.

**Is there a label or target associated with each instance?** If so, please provide a description.

Each instance represents an original image, along with its corresponding prompt and nine images generated by the stable diffusion model that potentially exhibit copyright infringement.

**Is any information missing from individual instances?** If so, please provide a description, explaining why this information is missing (e.g., because it was unavailable). This does not include intentionally removed information, but might include, e.g., redacted text.

No.

**Are relationships between individual instances made explicit (e.g., users' movie ratings, social network links)?** If so, please describe how these relationships are made explicit.

There is no explicit correlation between individual instances.

**Are there recommended data splits (e.g., training, development/validation, testing)?** If so, please provide a description of these splits, explaining the rationale behind them.

No.

**Are there any errors, sources of noise, or redundancies in the dataset?** If so, please provide a description.

No, there are no errors, sources of noise, or redundancies in the dataset.

**Is the dataset self-contained, or does it link to or otherwise rely on external resources (e.g., websites, tweets, other datasets)?** If it links to or relies on external resources, a) are there guarantees that they will exist, and remain constant, over time; b) are there official archival versions of the complete dataset (i.e., including the external resources as they existed at the time the dataset was created); c) are there any restrictions (e.g., licenses, fees) associated with any of the external resources that might apply to a future user? Please provide descriptions of all external resources and any restrictions associated with them, as well as links or other access points, as appropriate.

The dataset is self-contained.

**Does the dataset contain data that might be considered confidential (e.g., data that is protected by legal privilege or by doctor-patient confidentiality, data that includes the content of individuals non-public communications)?** If so, please provide a description.

No, the dataset does not contain data that might be considered confidential, such as information protected by legal privilege, doctor-patient confidentiality, or the content of individuals' non-public communications.

**Does the dataset contain data that, if viewed directly, might be offensive, insulting, threatening, or might otherwise cause anxiety?** If so, please describe why.

No.

**Does the dataset relate to people?** If not, you may skip the remaining questions in this section.

Yes

**Does the dataset identify any subpopulations (e.g., by age, gender)?** If so, please describe how these subpopulations are identified and provide a description of their respective distributions within the dataset.

The dataset primarily comprises four categories: Style, Portrait, Artistic Creation Figure, and Licensed Illustration.

**Is it possible to identify individuals (i.e., one or more natural persons), either directly or indirectly (i.e., in combination with other data) from the dataset?** If so, please describe how.

Yes, portraits are also part of copyright, and therefore, we have included a subset of celebrity portraits in the dataset.

**Does the dataset contain data that might be considered sensitive in any way (e.g., data that reveals racial or ethnic origins, sexual orientations, religious beliefs, political opinions or union memberships, or locations; financial or health data; biometric or genetic data; forms of government identification, such as social security numbers; criminal history)?** If so, please provide a description.

No.

**Any other comments?**

| Collection Process |
|:---:|

**How was the data associated with each instance acquired?** Was the data directly observable (e.g., raw text, movie ratings), reported by subjects (e.g., survey responses), or indirectly inferred/derived from other data (e.g., part-of-speech tags, model-based guesses for age or language)? If data was reported by subjects or indirectly inferred/derived from other data, was the data validated/verified? If so, please describe how.

The data was directly observable.

**What mechanisms or procedures were used to collect the data (e.g., hardware apparatus or sensor, manual human curation, software program, software API)?** How were these mechanisms or procedures validated?

We propose a pipeline to coordinate CLIP, ChatGPT, and diffusion models to generate a dataset that contains anchor images, corresponding prompts, and images generated by text-to-image models, reflecting the potential abuses of copyright. Initially, we collect a set of images that potentially contain copyrighted content, which serves as anchor images. Subsequently, these images are fed into the CLIP-interrogator, allowing us to obtain prompts that correspond to each anchor image. Finally, the prompts are used as input for the stable diffusion model, resulting in the generation of images by the stable diffusion model. Through manual comparisons, we assess whether there is evidence of copyright infringement in terms of style and semantics between the anchor images and the generated images. Ultimately, the anchor images, their corresponding prompts, and the images generated by the stable diffusion model constitute the core components of our dataset.

**If the dataset is a sample from a larger set, what was the sampling strategy (e.g., deterministic, probabilistic with specific sampling probabilities)?**

No.

**Who was involved in the data collection process (e.g., students, crowdworkers, contractors) and how were they compensated (e.g., how much were crowdworkers paid)?**

The dataset was primarily curated with contributions from the first three authors listed in the author list.

**Over what timeframe was the data collected? Does this timeframe match the creation timeframe of the data associated with the instances (e.g., recent crawl of old news articles)?** If not, please describe the timeframe in which the data associated with the instances was created.

The data was collected within the past three months.

**Were any ethical review processes conducted (e.g., by an institutional review board)?** If so, please provide a description of these review processes, including the outcomes, as well as a link or other access point to any supporting documentation.

No.

**Does the dataset relate to people?** If not, you may skip the remaining questions in this section.

Yes.

**Did you collect the data from the individuals in question directly, or obtain it via third parties or other sources (e.g., websites)?**

We obtained portrait information of public figures from Wikipedia.

**Were the individuals in question notified about the data collection?** If so, please describe (or show with screenshots or other information) how notice was provided, and provide a link or other access point to, or otherwise reproduce, the exact language of the notification itself.

Our images are sourced from Wikipedia, where the images are available for non-commercial or educational use.

**Did the individuals in question consent to the collection and use of their data?** If so, please describe (or show with screenshots or other information) how consent was requested and provided, and provide a link or other access point to, or otherwise reproduce, the exact language to which the individuals consented.

Our images are sourced from Wikipedia, where the images are available for non-commercial or educational use.

**If consent was obtained, were the consenting individuals provided with a mechanism to revoke their consent in the future or for certain uses?** If so, please provide a description, as well as a link or other access point to the mechanism (if appropriate).

We will provide our contact information on the release page of the website. In the event of any potential copyright infringement, we will promptly assess the situation, and if found to be valid, we will take immediate action to remove the corresponding data.

**Has an analysis of the potential impact of the dataset and its use on data subjects (e.g., a data protection impact analysis) been conducted?** If so, please provide a description of this analysis, including the outcomes, as well as a link or other access point to any supporting documentation.

We assert that our data is unlikely to cause potential negative impacts.

**Any other comments?**

---

| Preprocessing/cleaning/labeling |
| :---: |

**Was any preprocessing/cleaning/labeling of the data done (e.g., discretization or bucketing, tokenization, part-of-speech tagging, SIFT feature extraction, removal of instances, processing of missing values)?** If so, please provide a description. If not, you may skip the remainder of the questions in this section.

No.

**Was the "raw" data saved in addition to the preprocessed/cleaned/labeled data (e.g., to support unanticipated future uses)?** If so, please provide a link or other access point to the "raw" data.

**Is the software used to preprocess/clean/label the instances available?** If so, please provide a link or other access point.

**Any other comments?**

| Uses |
| :---: |

**Has the dataset been used for any tasks already?** If so, please provide a description.

This dataset is utilized for evaluating the efficacy of unlearning methods applied to stable diffusion.

**Is there a repository that links to any or all papers or systems that use the dataset?** If so, please provide a link or other access point.

No.

**What (other) tasks could the dataset be used for?**

This dataset can also be utilized to assist in determining whether copyright infringement has occurred.

**Is there anything about the composition of the dataset or the way it was collected and preprocessed/cleaned/labeled that might impact future uses?** For example, is there anything that a future user might need to know to avoid uses that could result in unfair treatment of individuals or groups (e.g., stereotyping, quality of service issues) or other undesirable harms (e.g., financial harms, legal risks) If so, please provide a description. Is there anything a future user could do to mitigate these undesirable harms?

No.

**Are there tasks for which the dataset should not be used?** If so, please provide a description.

No.

**Any other comments?**

| Distribution |
| :---: |

**Will the dataset be distributed to third parties outside of the entity (e.g., company, institution, organization) on behalf of which the dataset was created?** If so, please provide a description.

No.

**How will the dataset will be distributed (e.g., tarball on website, API, GitHub)** Does the dataset have a digital object identifier (DOI)?

We will release this dataset in github.

**When will the dataset be distributed?**

We plan to release our dataset upon the paper entering the review stage.

**Will the dataset be distributed under a copyright or other intellectual property (IP) license, and/or under applicable terms of use (ToU)?** If so, please describe this license and/or ToU, and provide a link or other access point to, or otherwise reproduce, any relevant licensing terms or ToU, as well as any fees associated with these restrictions.

The dataset is available for non-commercial or educational use.

**Have any third parties imposed IP-based or other restrictions on the data associated with the instances?** If so, please describe these restrictions, and provide a link or other access point to, or otherwise reproduce, any relevant licensing terms, as well as any fees associated with these restrictions.

The dataset is available for non-commercial or educational use.

**Do any export controls or other regulatory restrictions apply to the dataset or to individual instances?** If so, please describe these restrictions, and provide a link or other access point to, or otherwise reproduce, any supporting documentation.

The dataset is available for non-commercial or educational use.

**Any other comments?**

| Maintenance |
|:---:|

**Who will be supporting/hosting/maintaining the dataset?**

The researcher in this project.

**How can the owner/curator/manager of the dataset be contacted (e.g., email address)?**

We have provided the contact information of the dataset creators on our GitHub website.

**Is there an erratum?** If so, please provide a link or other access point.

No.

**Will the dataset be updated (e.g., to correct labeling errors, add new instances, delete instances)?** If so, please describe how often, by whom, and how updates will be communicated to users (e.g., mailing list, GitHub)?

We will release and update our dataset on GitHub, with a monthly update frequency.

**If the dataset relates to people, are there applicable limits on the retention of the data associated with the instances (e.g., were individuals in question told that their data would be retained for a fixed period of time and then deleted)?** If so, please describe these limits and explain how they will be enforced.

If our images infringe upon individuals' portrait rights, we will promptly remove the corresponding data after verification.

**Will older versions of the dataset continue to be supported/hosted/maintained?** If so, please describe how. If not, please describe how its obsolescence will be communicated to users.

 With each update, we incorporate the changes based on the original dataset, ensuring that previous versions of the data are preserved.

**If others want to extend/augment/build on/contribute to the dataset, is there a mechanism for them to do so?** If so, please provide a description. Will these contributions be validated/verified? If so, please describe how. If not, why not? Is there a process for communicating/distributing these contributions to other users? If so, please provide a description.

On the dataset's release page, we have provided corresponding links with the aim of encouraging collaborative expansion of the dataset and fostering the protection of copyright information.

**Any other comments?**