# OpenReview forum: "A Dataset and Benchmark for Copyright Protection from Text-to-Image Diffusion Models"
_ICLR.cc/2024/Conference — ICLR 2024 Conference Withdrawn Submission_

### Official Review · Reviewer_rnSG · 2023-10-24

**Soundness:** 1 poor
**Presentation:** 3 good
**Contribution:** 3 good
**Rating:** 3
**Confidence:** 4

**Summary:**

This paper aims on the problem of copyright.  It proposes two baselines to forget the specific concept or style to protect copyright, gradient ascent and pruning. They are proved to be more effective than previous methods for style forgetting and portrait forgetting.

Furthermore, It propose a dataset to test the forgetting methods. The dataset has four categories: "Style of Artist" includes images showcasing the unique artistic style of various artists, encompassing distinctive brushstrokes, lines, colors, and compositions found in their paintings. "Portrait of Celebrity" comprises of portrait images of notable individuals. "Artistic Creation Figure" consists of virtual cartoon and character images found in artistic and literary works. Lastly, "Licensed Illustration" encompasses artist illustrations that are protected by copyright.

**Strengths:**

The proposed two baselines for forgetting seems simple yet effective

**Weaknesses:**

1. This paper aims at a too big topic. The proposed 4 categories themselves are too vague. For example, *style*: art is still developing, and new art styles come out every day. Combining two art styles sometimes can be a new art style. How to define "combination" itself can be a copyright question; *portrait*: it is hard to use facial features to define the identity as many people look similar, such as twins, siblings. How similar can be considered as the same person? They might not be a problem when investigating "forgetting" methods, but it is a big problem when it comes to copyright. Furthermore, how to define "infringingly similar" itself is a very big problem, let alone copyright.
2. Although in Section 4.2, the dataset is divided into 4 parts: Style, Portrait, Artistic Creation Figure, and Licensed Illustration, the baselines are not tested in the four parts.
3. In the semantic metric, the cosine similarity is changed to MSE for CLIP scores. "easier to observe changes" is not a convincing reason. There is neither qualitative or quantitative experiment demonstrating it.
4. Although the definition of anchor images can be inferred from the paper, but there is no official definition of it in the paper.
5. In Figure 6, the unlearning methods totally break the model, generating meaningless things.

**Questions:**

1. Why take square in Equation 5? Both Semantic Metric and Style Metric are MSE, which are non-negative values.
2. What is the Semantic Metric and Style Metric are MSE for each experiment? Despite being more complicated, I believe analyzing two metrics can be more interesting. I would suggest to add both along with the proposed CM.
3. In Table 2, why FIDs are the same for Cartoon, Portrait, and Wikiart?
4. In Table 3, why the FID decreases after gradient ascent?

**Details Of Ethics Concerns:**

This paper aims on a really big, important, and sensitive problem: copyright of images from the text-to-image generative models. If the paper is not in a good enough format, it can be abused, considering the large related industry already existing in the world. Yet it is not good enough considering the weakness listed above.

---

### Official Review · Reviewer_oagV · 2023-11-03

**Soundness:** 3 good
**Presentation:** 3 good
**Contribution:** 3 good
**Rating:** 5
**Confidence:** 5

**Summary:**

This is an interesting work, which tackles the challenge of copyright infringement by text-to-image (stable diffusion) techniques. It highlights the absence of comprehensive studies, datasets, and standardized metrics in this domain. To address these gaps, the authors present the first dataset and benchmark for evaluating potential copyright abuses by SD, using a combination of CLIP, ChatGPT, and diffusion models to generate a relevant dataset and propose new evaluation metrics. These contributions aim to provide a foundation for future research on effective copyright protection in AIGC.

**Strengths:**

The strengths of this work is summarized below:
1. The creation of a specialized dataset that simulates potential abuses of copyright offers a tailored benchmark for testing and validating the proposed metrics. The dataset encompasses a variety of image types, including artistic styles, portraits, and animated characters, making it a versatile tool for research and application.
2. The benchmark itself is very novel and serves as a potential good pipeline for plagirism evaluation.

**Weaknesses:**

The weaknesses of this work come from the following aspects:
1. I think the main concern of this work is the lack of methodological novelty. Based on my understanding and my knowledge, the entire Sec. 3 is already well-known. It seems Sec. 4 does not incorperate too much novel methods. The main novelty of this work comes only from the dataset and the benchmark.
2. The lack of extensive study on the unlearning methods make this paper an incomplete work. For example, there are a lot of machine unlearning methods for concept-forgetting, and I listed them below. However, none of the methods is considered in this work. Without the results of the following methods, the reviewer believes this is far from a benchmark.
    > [1] Erasing Concepts from Diffusion Models

    > [2] Forget-Me-Not: Learning to Forget in Text-to-Image Diffusion Models

    > [3] Ablating Concepts in Text-to-Image Diffusion Models

    > [4] Unified Concept Editing in Diffusion Models

    > [5] Selective Amnesia: A Continual Learning Approach to Forgetting in Deep Generative Models

    > [6] Safe Latent Diffusion: Mitigating Inappropriate Degeneration in Diffusion Models

    > [7] SalUn: Empowering Machine Unlearning via Gradient-based Weight Saliency in Both Image Classification and Generation
3. I think Figure 3 is not informative enough, some key elements are missing, such as the color bar units. Also, it seems that there are some areas that have even larger magnitude than the diagnal line, is this normal? Finally, what if the semantic is copyrighted but the style is not (or reverse), will the CPDM metric not informative enough in this case?
4. Typo: in page 4 paragraph "Style metric":  inGatys et al. (2015) ->  in Gatys et al. (2015)

**Questions:**

I do not have additional questions and please see my comments in the Weakness column.

---

### Official Review · Reviewer_G9qh · 2023-11-07

**Soundness:** 3 good
**Presentation:** 3 good
**Contribution:** 3 good
**Rating:** 6
**Confidence:** 4

**Summary:**

The authors addressed the issue of copyright protection from text-to-image diffusion models. They have proposed a dataset for this task and performed benchmarking.

**Strengths:**

1. The task is novel and interesting. Content protection from diffusion model generated images are respectively less investigated and are potentially useful for copyright protection.
2. The problem setup, benchmarking and flow of the paper is clear and well-written.

**Weaknesses:**

1. Why CPDM metric is squared of two losses? Why not any other powers or any other combinations?
2. Can you provide some justification of defining the style metric?
3. Can there be any metric using content-style Disentanglement [ref], which is known to be editing dimensions of images?
4.  What will be the used cases for this unlearning task? Can you provide any practical test case ?
5. Can the unlearning be transferable? E.g., Can a portrait image be unlearned to be a cartoon image? Can this type of style transfer be done across categories in the dataset?

**Questions:**

Please check weakness